# SCALED NEURAL MULTIPLICATIVE MODEL FOR TRACTABLE OPTIMIZATION

## ABSTRACT

Challenging decision problems in retail and beyond are often solved using the predict-then-optimize paradigm. An initial effort to develop and parameterize a model of an uncertain environment is followed by a separate effort to identify the best possible solution of an optimization problem. Linear models are often used to ensure optimization problems are tractable. Remarkably accurate Deep Neural Network (DNN) models have recently been developed for various prediction tasks. Such models have been shown to scale to large datasets without loss of accuracy and with good computational performance. It can, however, be challenging to formulate tractable optimization problems based on DNN models. In this work we consider the problem of shelf space allocation for retail stores using DNN models. We highlight the trade-off between predictive performance and the tractability of optimization problems. We introduce a Scaled Neural Multiplicative Model (SNMM) with shape constraints for demand learning that leads to a tractable optimization formulation. Although, this work focuses on a specific application, the formulation of the models are general enough such that they can be extended to many real world applications.

## 1 INTRODUCTION

The predict-then-optimize framework is ubiquitous in applied research. A predictive model is first developed to approximate the true dynamics of a system under consideration to a given level of accuracy. A mathematical programming formulation based on the predictive model is then used to help researchers identify optimal policies for challenging real-world decision problems. Despite the fact that predictive models that are estimated based on the historical data need not be causal, there are recent works that shed some light on the principles behind this approach (Bertsimas & Kallus, 2020).

An alternative to the predict-then-optimize framework is integrating the two stages, predicting and optimizing at the same time, by utilizing a specific loss function in prediction models (Elmachtoub & Grigas, 2022). For example, model-free Reinforcement learning (RL) approaches explore their environment while also exploiting it to make optimal decisions.

Big-box retailers have thousands of stores located across the country. They use data-driven decision to optimize operations; e.g, assortment planning, pricing, and supply chain optimization. They work on problems related to shelf space allocation, making the best use of limited space in stores. Space planning for apparel is particularly challenging, due to the different shapes and sizes of the merchandise and the temporal shifts in brand importance.

The problem that motivated our work in this area involves deciding how much space in terms of *fixtures* to assign to each of several *brands* within a *category* of products for sale. These problems are typically solved at the *department* or *store* level. Products are arranged into departments and categories; for example, 'activewear tops' is a category within the women's apparel department. In this work, we introduce a novel approach based on the predict-then-optimize framework for solving a shelf space allocation problem.

In particular our contributions are as follows

- We begin by identifying certain key characteristics of relevant real-world data that make modeling challenging.

- We discuss predictive model selection considering the tractability of resulting optimization problem formulations and focus on the convexity of the formulations.

- We propose multiplicative models and establish conditions for tractability of the optimization. We also discuss why family of linear models are not suitable for the given application.

- We hypothesize that it is possible to convert an intractable optimization to a tractable one without loss of prediction accuracy. We achieve this via a Scaled Neural Multiplicative Model (SNMM) and demonstrate that our proposed model performs well relative to alternative models.

## 2 RELATED WORK

The relationship between the shelf space allocated to a product and that product's sales has been studied extensively (Bianchi-Aguiar et al., 2021; Hübner & Kuhn, 2012; Karampatsa et al., 2017). Analyses often focus on incremental returns, estimating the *space elasticity*. Under the assumption of diminishing returns to scale, the relationship between shelf space and sales can be modeled using a concave function (Curhan, 1972; Eisend, 2014). The assumption of diminishing returns makes intuitive sense and has been widely used in production functions (Gopalswamy & Uzsoy, 2019)(Aigner & Chu, 1968); the incremental gain in sales by adding more space to showcase a product decreases as the space allocated increases. This assumption can also have a side benefit of making shelf space allocation optimization problems easier to solve.

**Optimization:** There have been a number of scientific articles focused on the assortment optimization problem; a problem related to the one we consider in this work. Retailers solve this problem to determine which products to sell. Shelf space in stores is often the most prominent constraint. Kök & Fisher (2007), Yücel et al. (2009), Lo (2019), and numerous others focus on developing optimization frameworks that include complicated consumer choice models. This allows the authors to select the optimal product mix accounting for the fact that some consumers will substitute one product for another depending on what is and is not sold by a retailer. Our problem is somewhat different in that we are not selecting specific products to sell, but rather deciding which brands to carry and how much space to allocate to these brands. These brands will offer products that are relatively unique within a store.

Hübner & Kuhn (2011) points out the relationships between space allocated, consumer demand, and inventory costs. Assuming a fixed amount of space available, a retailer can choose to offer fewer facings of a more diverse set of products to increase consumer interest. This will, however, also increase inventory holding and replenishment costs. There will be increasing demands placed on store labor. Ryzin & Mahajan (1999) came to a similar conclusion earlier looking specifically at apparel. Our problem is, again, somewhat different. Brand managers are responsible for selecting product mix within brands sold in specific stores, managing inventory costs and the tradeoff between such costs and expected sales or profits.

**Shape-Constrained Models:** An integral part of our prediction model that makes the optimization tractable is shape constraints. These are models that are constrained to be monotonic, convex, concave or non-negative among others. Shape constraints provide effective regularization, reducing the chance that noisy training data or adversarial examples produce a model that does not behave as expected. These impose strong priors on the data and can be used effectively to produce well behaved structured prediction models. It is this property that we enforce in our prediction model to yield a tractable optimization. Shape constraints on neural networks have been studied in different applications (Gupta et al., 2018). Most of the prior works consider GAMs (Generalized Additive Models) and neural networks separately. In our work, we combine the strength of neural networks and the simplicity of GAMs and propose a Scaled Neural Multiplicative model that can model concave, convex or monotone constraints effectively.

**Differentiable Optimization:** Recent works have introduced classes of deep learning models where certain layers involves solving optimization problems (Donti et al., 2017; Agrawal et al., 2019; Wilder et al., 2019). In these differentiable optimization architectures, backpropagation is based on a loss function that reflects the decision problem(s). The goal is to directly learn a policy that performs

well on the optimization problems under the true distributions of uncertain parameters. Donti et al. (2017) demonstrate that this approach can outperform "both traditional modeling and purely black-box policy optimization approaches" on sample stochastic optimization problems. Agrawal et al. (2019) introduce a methodology for problems involving certain optimization problems they call disciplined convex programs. Wilder et al. (2019) introduce methods for linear and submodular maximization combinatorial optimization problems.

## 3 PRELIMINARIES

Given a closed set $\mathbb{X}$ and function $F_\theta$ that is parameterized by $\theta$, decision making problem is to find the solution to the optimization problem given below

$$\boldsymbol{x}^* = \arg\max_{\boldsymbol{x} \in \mathbb{X}} F_{\boldsymbol{\theta}}(\boldsymbol{x}) \tag{1}$$

The complexity of the underlying optimization model is based on the form of the function $F_\theta$, the objective function, and the definition of the set $\mathbb{X}$, the constraints. Further, the problem is generally tractable only for concave functions with respect to $\boldsymbol{x}$ (including affine).

Although, more generic functional forms can be incorporated using Mixed Integer Programming (MIP) with piecewise linear approximations (Vielma et al., 2010)(Gopalswamy et al., 2019), the solution time for MIPs in general does not scale well for large problems. By tractability, we refer to the class of optimization problems that can be solved in polynomial time.

A linear functional form for the objective function leads to a relatively simple convex (linear) optimization problem if the set $\mathbb{X}$ is a convex polytope. In this case, the prediction model must be linear with respect to the decisoptimization variable $\boldsymbol{x}$. The $\boldsymbol{c}$ model parameters are coefficients in a linear function of $\boldsymbol{x}$ while the rest of the features $\mathbb{P}\backslash\boldsymbol{x}$ in the prediction model do not have any restrictions. The predictive model can be a generic neural network, for instance. Note that features other than $\boldsymbol{x}$ are irrelevant to the decision problem.

$$F_{\boldsymbol{\theta}}(\boldsymbol{x}) = \boldsymbol{c}^t \boldsymbol{x} + \phi(\mathbb{P}\backslash\boldsymbol{x})$$

A multiplicative model form can be a better alternative to a linear model form for two reasons: *(i) it can handle heterogeneity in variance w.r.t features in $\mathbb{X}$ and (ii) other features in $\mathbb{P}$ can play a role in determining the optimal solution to equation 1*. A general multiplicative model is given by

$$F_{\boldsymbol{\theta}}(\boldsymbol{x}) = \prod_i x_i^{\beta_i} \phi(\mathbb{P}\backslash x_i) \tag{2}$$

If $\sum_i \beta_i = 1$ and $\phi(\mathbb{P}\backslash x_i) \in \mathbb{R}^+$, then our objective function is convex. Such conditions can be difficult to enforce on a general neural network model without explicit architectural design (Amos et al., 2017). Further, such models can be difficult to interpret and communicate to business stakeholders in general. To alleviate the above problems, we consider the following multiplicative form

$$F_{\boldsymbol{\theta}}(\boldsymbol{x}) = \sum_i x_i^{\beta_i} \prod_{p \in \mathbb{P}\backslash x_i} \phi_p(w_i^p) \tag{3}$$

where $w_i^p$ is the feature $p$ that depends on the index $i$ of the optimization variable. For example, variable $x_i$ could represent space for brand $i$ whereas $w_p^i$ could be description of brand $i$ embedded using a language model. Details on how to constraint 3 to enforce convexity (concavity) will be discussed in the next section.

## 4 PROBLEM FORMULATION

We discuss a specific instance of the optimization problem based on the predictive model $F_\theta$ that captures the relationship between sales and features. The variable of interest in the optimization, *fixture count*, defines the shelf space in a store. This work specifically considers the problem of finding optimal space allocation for each brand-category pair for a given department in each store to maximize revenue.

The tractability of the above formulation depends on the set $\mathbb{X}$ and the parametric function $F_\theta$. In this work we will consider a family of functions from Generalized linear models (GLM) such as additive, multiplicative as well as DNNs that extend GAMs (Agarwal et al., 2021). We will analyze the models from the perspective of flexibility and convexity with respect to the fixture count feature

We approach this problem in two stages -

- Learn the functional form of $F_{\boldsymbol{\theta}}(\boldsymbol{x})$ through demand learning.
- Use this relationship to optimize for space across all stores.

## 4.1 DATA SPARSITY

A major challenge in pushing research from academia to application rests with the data quality. It is non trivial to achieve the appropriate functional form or relationship that captures the expected model behavior without loss in performance. In the retail space in particular, we have found that the domains of the functions we would like to use are not fully observed in the historical data. This is due to the challenges in experimentation in physical retail stores at the granularity needed for making decisions. For example, in apparel there are a large number of items sold, while the sales traffic at each item level is very sparse. Ideally, one would be interested in estimating the space elasticity of item $i$ for time period $t$. This estimation problem is challenging due to sparsity in data and non-dynamic space allocation in retail stores (i.e, space allocation for an item or category usually stays the same for a quarter to offset high labor costs involved). We consider models that have a single coefficient for fixture count. Further, the category and brand have multiplicative effects in the models we consider. This way, we are able to estimate the space elasticity by using data across different products and also estimate total effect for any given brand and category.

## 4.2 GENERALIZED LINEAR MODELS

**Linear Fixed Effects Model:** A baseline approach involves fitting a linear regression model where $y$, the \$ sales amount, is a linear combination of $x$ and $d_i$, the features. Here, $x$ denotes the fixture count variable and $d_i$ denotes other features like brand, category and department.

$$y = \beta_0 + \beta_1 x + \sum_i \xi_i d_i \tag{4}$$

When used as an objective function with $x$ as the decision variable, the linear model has constant space elasticity across all brand-category pairs and the effect of other features other than fixture count becomes irrelevant to the optimization.

**Log-Log Model:** A log-log model ensures that the effect of fixture count on sales is positive. Moreover, unlike a simple linear model, the coefficients of log-log model have a multiplicative effect on the independent variable. The multiplicative effect of coefficients here ensures that the effect of different features like brand, category and department are directly related to the fixture count variable. These variations across brand, category and department learned from the historical data allows the optimization model to identify the right weights associated with each fixture count variable and assign space accordingly.

$$\log y = \beta_0 + \beta_1 \log x + \sum_i \xi_i d_i$$
$$y = e^{\beta_0} x^{\beta_1} e^{\sum_i \xi_i d_i}$$
$$y = F(\boldsymbol{d}) x^{\beta_1} \tag{5}$$

The log-log model is convex when $\beta_1 > 1$ or $\beta_1 < 0$. In our motivating example, these would be the cases where allocating additional space to a brand would either reduce sales or increase sales at a faster rate than before (increasing returns to scale). Neither of these cases seem realistic.

## 4.3 DEEP NEURAL NETWORKS

**Neural Additive Models (NAM):** Deep neural networks have been successful in prediction tasks. We consider a special form of neural networks that are additive in nature (Agarwal et al., 2021).

NAMs learn a linear combination of networks, each of which attend to a single input feature: each $f_i$ in 6 is parameterized by a neural network. NAMs are explainable, elegant and easy to understand models.

$$y = \beta + f_1(x_1) + f_2(x_2) + \cdots + f_K(x_K) \tag{6}$$

**Neural Multiplicative Models (NMM):** Similar to the multiplicative linear models, we consider multiplicative form for the neural additive model using the log transformation of the dependent variable $y$ and fixture count $x$

$$y = e^\beta e^{f(\log x)} e^{\sum_i f_i^d(d_i)} \tag{7}$$

Equation 7 in general will not lead to a tractable optimization problem. The functional form considered in Agarwal et al. (2021) involves EXU layer $g(x) = e^w(x - b)$ and linear layers. We consider a linear functional form of $f$ that can be modeled via linear layers. To improve learning and expressiveness we use hidden dimension of 100, thus forming a total of 2 linear layers.

$$y = e^\beta x^{\boldsymbol{w}_2'\boldsymbol{w}_1} e^{\boldsymbol{w}_2'\boldsymbol{b}_1 + b_2} e^{\sum_i f_i^d(d_i)} \tag{8}$$

The parameters of Equation 8 can be constrained to produce a concave function well suited for use as an objective function in a subsequent optimization problem. The constraint $0 \leq \boldsymbol{w}_2'\boldsymbol{w}_1 \leq 1$, can be modeled by considering $\sigma(\boldsymbol{w}_2), \sigma(\boldsymbol{w}_1)$ instead $\boldsymbol{w}_2, \boldsymbol{w}_1$, where $\sigma(\boldsymbol{x}) = \left\{ \frac{\exp x_i}{\sum_i \exp x_i} \right\}_{i=1}^n$. It can be shown that $0 \leq \sigma(\boldsymbol{w}_2)'\sigma(\boldsymbol{w}_1) \leq 1$.

**Lemma 4.1.** *For any $x, y \in \Delta$ where $\Delta = \{w \in \mathcal{R}_+^n | \sum_i w_i = 1\}$, we have $0 \leq \sum_i x_i y_i \leq 1$*

*Proof.* In appendix. $\square$

**Scaled Neural Multiplicative Models (SNMM):** Generally, constraining neural network models leads to reduction in their capacity. The direct consequence being the inability to be a universal function approximator. We propose a NMM model where the scaling of the feature $x$ before the log transformation can be learned in an end-to-end fashion.

$$y = e^\beta z^{\sigma(\boldsymbol{w}_2)'\sigma(\boldsymbol{w}_1)} e^{\sigma(\boldsymbol{w}_2)'\boldsymbol{b}_1 + b_2} e^{\sum_i f_i^d(d_i)} \tag{9a}$$

$$z = 1 + \max(0, \boldsymbol{s}_2'\boldsymbol{s}_1 x + \boldsymbol{s}_2'\boldsymbol{t}_1 + t_2) \tag{9b}$$

Equation 9 is tractable with respect to the optimization when $\boldsymbol{s}_2'\boldsymbol{s}_1 x + \boldsymbol{s}_2'\boldsymbol{t}_1 + t_2 \geq 0$. Although, $\max$ operator cannot be modeled without mixed integer program, we noticed that the above condition for tractability holds on the experiments we carried out with different number of stores. In our application, $y$ represents sales in dollars and $x$ represents fixture count. Equation 9 can be written for an explicit category $i$, brand $j$ and department $k$ as follows

$$y_{i,j,k} = a_{i,j,k} z_{i,j,k}^\gamma \tag{10a}$$

$$z_{i,j,k} = \boldsymbol{s}_2'\boldsymbol{s}_1 x_{i,j,k} + \boldsymbol{s}_2'\boldsymbol{t}_1 + t_2 \tag{10b}$$

$$a_{i,j,k} = e^\beta e^{\sigma(\boldsymbol{w}_2)'\boldsymbol{b}_1 + b_2} e^{f_i(d_i) + f_j(d_j) + f_k(d_k)} \tag{10c}$$

Note that $a_{i,j,k}$ is constant with respect to optimization variables.

### 4.4 OPTIMIZATION

After learning the functional form of $F_\theta$, we can formulate the problem for stage two as follows. Let the sets $\mathbb{I}$ and $\mathbb{K}$ denote the set of brand, category pairs $(i, j)$ and set of departments, respectively. The parameter $a_{i,j,k}$ is the coefficient of category $i$, brand $j$ and department $k$ based on the multiplicative model in 9. Let $x_{i,j,k}$ denote the variable fixture count for category $i$, brand $j$ and department $k$, with $l_{i,j,k}$ and $u_{i,j,k}$ as lower and upper bounds on the variable, respectively. Further, lets define parameters $s$ and $s_k$ and $\gamma$ as the total fixture count (across all departments), fixture count for a department $k$ and space elasticity, respectively.

We define the optimization with explicit index $(i, j)$ to indicate the dependency of the coefficients $a_{i,j,k}$ on brand, category and department encoding, even though it can be combined into a single index. The objective is to maximize sales revenue from equation 10. The explicit form of equation 10,

$f(x_{i,j,k})$, depends on the model of choice (SNMM in our case).

$$\max \sum_{k \in \mathbb{K}} \sum_{i,j \in \mathbb{I}} y_{i,j,k} \tag{11a}$$

$$s.t.$$

$$y_{i,j,k} = f(x_{i,j,k}), \forall i, j, k \tag{11b}$$

$$\sum_{i,j \in \mathbb{I}} x_{i,j,k} \leq s_k, \forall k \in \mathbb{K} \tag{11c}$$

$$\sum_{k \in \mathbb{K}} \sum_{i,j \in \mathbb{I}} x_{i,j,k} \leq s \tag{11d}$$

$$l_{i,j,k} \leq x_{i,j,k} \leq u_{i,j,k}, \forall i, j, k \tag{11e}$$

$$x_{i,j,k} \geq 0, \forall i, j, k \tag{11f}$$

## 5 EXPERIMENTS

We demonstrate the ability of the proposed model to handle shape constraints and predictive power with two sets of experiments. In the first experiment, we compare SNMM with different models from (Gupta et al., 2018) using real world datasets on their ability to handle constraints such as concavity with respect to the input. The models compared include: a "standard unconstrained DNN" (Gupta et al., 2018), partial convex shape-constrained neural networks (Amos et al., 2017), Random Tiny Lattices (RTLs) (Canini et al., 2016), and calibrated linear models (Gupta et al., 2016). See Gupta et al. (2018) for more details on these models. Our results regarding their performance are taken from this work. In the second experiment, we provide results on a proprietary dataset from a retailer with respect to modeling space elasticity.

### 5.1 SHAPE-CONSTRAINED BENCHMARK

We consider three publically available datasets and test the ability of SNMM to handle convexity and concavity constraints based on established benchmarks.

Table 1: Synthetic experiments: Car Sales and Puzzle Review

| **Car Sales** | | | **Puzzle Sales** | | |
|---|---|---|---|---|---|
| Model | Val. MSE | Test MSE | Model | Val. MSE | Test MSE |
| **SNMM** | **1923** | **8486** | SNMM | 3768 | 8457 |
| DNN | 2035 | 10931 | **DNN** | **2189** | **5652** |
| SCNN conv. | 2262 | 10613 | SCNN conc. | 2632 | 7931 |
| SCNN conv. decr. | 2442 | 10590 | SCNN conc. incr. | 2437 | 6927 |
| Cal Lin. decr. | 2271 | 10727 | RTL incr. | 4457 | 8838 |
| Cal Lin. conv. decr. | 2304 | 10593 | RTL all | 3543 | 8315 |
| | | | Cal Lin. incr. | 3589 | 8270 |
| | | | Cal Lin. all | 3617 | 8189 |

### 5.1.1 CAR SALES

This dataset is a 1-d problem with 109 training, 14 validation, and 32 test examples (www.kaggle.com/hsinha53/car-sales/data), for which we predict monthly car sales (in thousands) from the price (in thousands). We enforce a convexity constraint on the price variable with respect to the sales variable. The results can be seen in Table 1. SNMM performed relatively better compared to all models with the lowest Validation and Test MSE.

### 5.1.2 PUZZLE SALES FROM REVIEWS

The data consists of 3 features, 156 training, 169 validation, and 200 non-IID test examples (courtesy of Artifact Puzzles and available at `www.kaggle.com/dbahri/puzzles`), for which we predict the 6-month sales of different wooden jigsaw puzzles from three features based on its Amazon reviews: its average star rating, the number of reviews, and the average word count of its reviews. Here we assume a convexity constraint on the star rating, and concavity constraints on number of reviews and word count. Table 1 shows that for SNMM, even though the Validation MSE is highest compared to the rest of the models, the Test MSE remains within the range of the highest Test MSE. The results aren't as impressive as on the Car Sales problem, indicating that our assumptions regarding convexity and concavity are more limiting here. It is important to note that these assumptions are still necessary to develop a tractable optimization problem formulation.

Table 2: Synthetic experiments: Wine Quality

| Model | Wine Quality | |
| | Val. MSE | Test MSE |
| --- | --- | --- |
| SNMM | 6.95 | 6.98 |
| **DNN** | **4.91** | **4.79** |
| SCNN conc. | 5.96 | 7.22 |
| SCNN conc. incr. | 6.13 | 6.21 |
| RTL incr. | 4.96 | 4.85 |
| RTL conc. incr. | 4.96 | 4.83 |
| Cal Lin. incr. | 5.25 | 5.10 |
| Cal Lin. conc. incr. | 5.23 | 5.10 |

### 5.1.3 WINE ENTHUSIAST MAGAZINE REVIEWS

The data (61 features, 84,642 training, 12,092 validation, and 24,185 test examples; `www.kaggle.com/dbahri/wine-ratings`) consists of wine's quality measured in points [80, 100], price of the wine (the most important feature), country (21 Bools), and 39 Bool features based on the wine description from Wine Enthusiast Magazine. We constrain the price feature to be concave in order to predict the quality of wine. From Table 2, it can be seen that while DNN has the lowest error, the Test MSE for SNMM is within the range of the highest Test MSE across all models.

Table 3: Benchmark results for 5 stores

| Model | log scale | | | linear scale | | |
| | $R^2$ | MSE | MAE | $R^2$ | MSE | MAE |
| --- | --- | --- | --- | --- | --- | --- |
| GLM-L | - | - | - | $0.623 \pm 0.01$ | $61692 \pm 2214$ | $126 \pm 0.67$ |
| GLM-M | $0.759 \pm 0.00$ | $0.531 \pm 0.00$ | $0.568 \pm 0.00$ | $0.619 \pm 0.01$ | $62398 \pm 1323$ | $96 \pm 0.64$ |
| GLM-CM | $0.736 \pm 0.00$ | $0.582 \pm 0.00$ | $0.603 \pm 0.02$ | $0.618 \pm 0.01$ | $62525 \pm 2202$ | $97 \pm 0.87$ |
| NAM | - | - | - | $0.684 \pm 0.01$ | $55809 \pm 2033$ | $100 \pm 1.90$ |
| NMM | $0.769 \pm 0.01$ | $0.506 \pm 0.01$ | $0.564 \pm 0.01$ | $0.567 \pm 0.01$ | $69923 \pm 1672$ | $95 \pm 1.46$ |
| NMM-L | $0.824 \pm 0.02$ | $0.388 \pm 0.05$ | $0.483 \pm 0.02$ | $0.738 \pm 0.06$ | $43917 \pm 1099$ | $80 \pm 5.44$ |
| NMM-CL | $0.638 \pm 0.01$ | $0.798 \pm 0.01$ | $0.724 \pm 0.01$ | $0.407 \pm 0.01$ | $96364 \pm 2771$ | $120 \pm 1.61$ |
| SNMM | $\mathbf{0.827 \pm 0.01}$ | $\mathbf{0.382 \pm 0.02}$ | $\mathbf{0.476 \pm 0.01}$ | $\mathbf{0.775 \pm 0.02}$ | $\mathbf{37830 \pm 2687}$ | $\mathbf{75 \pm 2.32}$ |

### 5.2 CASE STUDY ON REAL DATA

In this section, we perform tests on a random selection of stores across the country from one retailer. We first describe the dataset and the features considered. We will then compare eight different

models for predicting sales with the metrics given below.

$$R^2 = 1 - \frac{\|\boldsymbol{y} - \hat{\boldsymbol{y}}\|^2}{\|\boldsymbol{y} - \bar{\boldsymbol{y}}\|^2} \tag{12}$$

$$MSE = N^{-1}\|\boldsymbol{y} - \hat{\boldsymbol{y}}\|^2 \tag{13}$$

$$MAE = N^{-1}|\boldsymbol{y} - \hat{\boldsymbol{y}}| \tag{14}$$

We consider linear and multiplicative models with variants. The variants are explored in lieu of search for computationally tractable optimization model. Further, certain models offer better modeling capacity with respect to the solution space. The functional relationship between the space feature (variable in optimization) and sales defines the objective function. We expect the models to capture the concave relationship from the data, but it seldom happens in reality. Real-word data is noisy and unobserved confounders can affect the data. Further, retail data in general depends on exogenous uncertainty such as market dynamics, competitors, economic factors, etc. Explicit shape constraint is placed on the model such as non-negativity of sales and concavity of sales with respect to space feature.

Table 4: Benchmark results for 10 stores

| Model | log scale | | | linear scale | | |
| | $R^2$ | MSE | MAE | $R^2$ | MSE | MAE |
|---|---|---|---|---|---|---|
| GLM-L | - | - | - | $0.638 \pm 0.00$ | $56655 \pm 1378$ | $121 \pm 0.43$ |
| GLM-M | $0.753 \pm 0.00$ | $0.522 \pm 0.00$ | $0.564 \pm 0.00$ | $0.657 \pm 0.01$ | $53653 \pm 1465$ | $90 \pm 0.39$ |
| GLM-CM | $0.727 \pm 0.00$ | $0.577 \pm 0.00$ | $0.601 \pm 0.00$ | $0.616 \pm 0.01$ | $60084 \pm 1404$ | $92 \pm 0.46$ |
| NAM | - | - | - | $0.668 \pm 0.00$ | $53217 \pm 827$ | $98 \pm 1.09$ |
| NMM | $0.786 \pm 0.01$ | $0.452 \pm 0.01$ | $0.533 \pm 0.01$ | $0.622 \pm 0.01$ | $59894 \pm 2255$ | $86 \pm 1.52$ |
| NMM-L | $0.813 \pm 0.01$ | $0.396 \pm 0.02$ | $0.488 \pm 0.01$ | $0.707 \pm 0.01$ | $46438 \pm 2315$ | $78 \pm 2.20$ |
| NMM-CL | $0.648 \pm 0.02$ | $0.746 \pm 0.04$ | $0.7 \pm 0.03$ | $0.478 \pm 0.04$ | $82035 \pm 7789$ | $109 \pm 5.24$ |
| SNMM | $\mathbf{0.817 \pm 0.01}$ | $\mathbf{0.387 \pm 0.00}$ | $\mathbf{0.479 \pm 0.00}$ | $\mathbf{0.760 \pm 0.01}$ | $\mathbf{37143 \pm 1996}$ | $\mathbf{72 \pm 0.51}$ |

As discussed in sections 4.2 and 4.3, we enforce the constraints and report performance metrics. We present the metrics in both log scale and linear scale whenever applicable to offer complete comparison across all the models

The following models are used for the experiment:

- GLM-L: Classical linear regression model that assumes a linear relationship between the dependent and independent variables, Equation 4.

- GLM-M: Multiplicative model that takes a log transformation of dependent variable and space variable as in Equation 5.

- GLM-CM: Constrained GLM-M with space elasticity between 0 and 1.

- NAM: Neural Additive Model as described in Equation 6

- NMM: NAM that takes a log transformation of dependent variable and space variable as in Equation 7.

- NMM-L: NMM with linear layers for the space variable as in Equation 8

- NMM-CL: Constrained NMM-L with space elasticity between 0 and 1 using lemma 4.1

- SNMM: Proposed model in Equation 9

**Dataset:** The models consider historical space and sales data of Apparel departments from 2018 to 2022. We want to predict and optimize for space at the department-category-brand level, so we aggregate data to that level at a weekly granularity.

**Model Evaluation:** Numerical tests were carried out on a set of 5, 10 and 20 stores. The effectiveness of the models in sales prediction is evaluated using the metrics described above. The proposed model is compared with variety of different models and the results are shown in Tables 3,

Table 5: Data summary

| Feature | Description |
|---------|-------------|
| $ sales | Dependent variable representing the dollar sales amount |
| fixture count | Measure of space represented in a store |
| store no | Store number (categorical) |
| store area | Area of a store in sq. ft. |
| department no | Department number |
| category | Category description (vector embedding) |
| brand | Name of the brand (vector embedding) |
| demographic | Demographic data based on location of the store |
| income | Income data based on location of the store |
| time features | Time-based features based on the week of transaction |

Table 6: Benchmark results for 20 stores

| Model | log scale $R^2$ | MSE | MAE | linear scale $R^2$ | MSE | MAE |
|-------|-----------------|-----|-----|--------------------|-----|-----|
| GLM-L | - | - | - | $0.583 \pm 0.00$ | $71124 \pm 1943$ | $126 \pm 0.29$ |
| GLM-M | $0.742 \pm 0.00$ | $0.531 \pm 0.00$ | $0.568 \pm 0.00$ | $0.626 \pm 0.01$ | $63831 \pm 3224$ | $91 \pm 0.56$ |
| GLM-CM | $0.715 \pm 0.00$ | $0.587 \pm 0.00$ | $0.605 \pm 0.00$ | $0.575 \pm 0.00$ | $72437 \pm 1849$ | $92 \pm 0.50$ |
| NAM | - | - | - | $0.617 \pm 0.00$ | $65838 \pm 1475$ | $102 \pm 0.57$ |
| NMM | $0.793 \pm 0.01$ | $0.428 \pm 0.01$ | $0.516 \pm 0.01$ | $0.585 \pm 0.01$ | $72292 \pm 841$ | $84 \pm 0.35$ |
| NMM-L | $\mathbf{0.807 \pm 0.01}$ | $\mathbf{0.400 \pm 0.01}$ | $\mathbf{0.487 \pm 0.01}$ | $0.682 \pm 0.01$ | $55934 \pm 2198$ | $79 \pm 1.12$ |
| NMM-CL | $0.623 \pm 0.01$ | $0.78 \pm 0.03$ | $0.713 \pm 0.02$ | $0.352 \pm 0.02$ | $113907 \pm 4103$ | $113 \pm 2.49$ |
| SNMM | $\mathbf{0.806 \pm 0.02}$ | $\mathbf{0.399 \pm 0.03}$ | $\mathbf{0.486 \pm 0.02}$ | $\mathbf{0.716 \pm 0.01}$ | $\mathbf{48983 \pm 2228}$ | $\mathbf{74 \pm 2.56}$ |

4 and 6. The results are reported on the `test-split`. Models are tuned for hyper-parameters - `batch-size {32,64,128,256,512}`, `hidden-size {10,50,100}`.

SNMM model performs the best across all the scenarios; MSE and MAE for SNMM are the lowest across all the models. Comparable performance can be seen in NMM-L model where we enforce linearity of the space feature in log space. We highlight the benefit of enforcing structured constraints by comparison to several other models that are unconstrained. Although constrained models usually have lower capacity, SNMM is able to recover the performance lost due to constraints when compared to the constrained models and even improve on them. This clearly shows that the generalization ability of the shape constrained model is due to regularization with respect to the structure.

## 6 CONCLUSION AND FUTURE WORK

In this work, we presented a shelf space allocation problem for retail use case. A general framework of predict and optimize is discussed in detail with regards to trade-off between tractability of the optimization and prediction model accuracy. We discuss the need for multiplicative models and the difficulty in estimating space elasticity on a sparse dataset (most often this is the case in real world problems). To that extent, we propose a Scaled Neural Multiplicative Model (SNMM) that satisfies the conditions: non-negativity of sales and concavity with respect to the space feature. The optimization problem is formulated as a convex problem which can solved using convex solvers. Specifically, we use the power cone formulation provided by `cvxpy`. The model proposed in this work is general enough to extend to many other applications such as advertisement optimization, revenue maximization, etc. In future, we plan to explore these areas and enhance the model to handle different sets of constraints jointly.

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

# A  APPENDIX

*Proof.* **Lemma 4.1**

$$\left( \sum_i x_i y_i \right)^2 \leq \left( \sum_i x_i^2 \right)\left( \sum_i y_i^2 \right) \quad \text{By Cauchy Shwartz inequality}$$

$$\leq \left( \sum_i x_i \right)\left( \sum_i y_i \right) \quad \because x_i \leq 1, y_i \leq 1 \implies x_i^2 \leq x_i, y_i^2 \leq y_i$$

$$\sum_i x_i y_i \leq 1 \quad \because \sum_i x_i = \sum_i y_i = 1$$

Trivially, $\sum_i x_i y_i \geq 0$ since $x_i, y_i \geq 0 \ \forall i$ $\qquad\qquad\qquad\qquad\qquad\qquad\square$

