# OpenReview forum: "Scaled Neural Multiplicative Model for Tractable Optimization"
_ICLR.cc/2023/Conference — Submitted to ICLR 2023_

### Official Review · Reviewer_1jyf · 2022-10-24

**Confidence:** 3
**Correctness:** 2
**Technical Novelty And Significance:** 2
**Empirical Novelty And Significance:** 1
**Recommendation:** 1

**Clarity, Quality, Novelty And Reproducibility:**

**Novelty & Significance**

The novelty of the paper is not very well presented, and from what it stands now, seems quite limited. In particular,
1. The formulation targets a very specific optimization problem, which significantly limits the scope of the paper. It is important to justify the generality of the proposed method by either 1) reformulating the problem in a more general way and demonstrating how the method still fits, or 2) incorporating a discussion of how the proposed method generalizes beyond the specific problem formulation.
2. The proposed SNMM simply modifies the NMM by adding a scaler parameter for the input feature. This modification, however, seems to be minor and marginal. Furthermore, there may be many other ways that could be considered, e.g., by utilizing the composition of (monotonically increasing) convex functions, but are not discussed or compared in the paper. It is crucial to justify the significance of the proposed SNMM compared with potential variants.
3. The empirical evaluation of the proposed SNMM is not convincing. In particular, SNMM seems to work poorly on some evaluated setups (e.g., Table 2) and leads to marginal improvements on the real-world dataset (Tables 3, 4, 6), which makes its significance even more questionable. See “Reproducibility” for more details.


**Clarity & Quality**

The current presentation of the paper is not clear and sometimes confusing. It took me some time to figure out what problem the paper tries to address and how the paper goes toward it. There are also several unjustified claims and mathematical unclarities in the paper. I detail my confusion and questions below:
1. *The problem formulation is not clear*. The paper focuses on the specific problem of ​​shelf space allocation, but the specific description of the paper and detailed problem formulation is not presented beforehand, it would be much clear if the problem is well presented at the start of Section 4 or 3.
2. *The proposed method and the contributions are not clear*.
    * The key issue that the proposed SNMM aims to address, which (to my understanding) is to increase the expressivity of the NMM under the introduced constraint, needs to be very clearly stated out.
    * In the paragraph that introduces SNMM, it is loosely claimed that “Generally, constraining neural network models leads to reduction in their capacity” which is not necessarily true and needs further discussion or empirical evidence. It is also important to discuss how the proposed SNMM breaks through the expressivity bottleneck under the constraint to justify the proposed method.
    * In Eq. (9b), the purpose for introducing the $\max$ operator is not explained.
    * In Sec 4.4, how the optimization problem is solved is not discussed.
3. *Some statements are not well explained or justified*. Here are a few:
    * In Eq. (8), the two linear layers are composed without any linearity which still falls into the linear family. How does it increase the expressivity as claimed above?
    * At the start of Sec 3.2, the two reasons for the advantages of multiplicative models over linear models are not sufficiently supported. More explanations or some references are needed.
4. *Missing mathematical definitions*. Here are a few:
    * In Sec 3.1, the equation for $F_\theta(x)$, both $\phi$ and $\mathbb{P}$ are not defined.
    * In Eq. (3), the notation $\phi_p(w_i^p)$ is confusing to me, what doe $p$ mean as sub-script and super-script respectively?
    * In Eq. (14), the notation $N$ is not defined.


**Reproducibility**

The reproducibility of the experimental results is poor. In particular, in Sec 5.1, the experimental setups (e.g., how the models are trained and how the optimization problem is solved) and even the baselines considered in Table 1 & 2 are not introduced. In Sec 5.2, the dataset used seems not publicly available, but the dataset descriptions are not detailed enough for reproducibility. There are no further experimental details included for reproducing the results in the paper.


**Details Of Ethics Concerns:**

In Sec 5.1.1-5.1.3, the starting sentences for introducing the datasets are all copied from Gupta et al, 2018, which may be considered a violation of academic integrity.

**Strength And Weaknesses:**

**Strength**
* The paper considers an optimization problem that is practically relevant. The collected real-world dataset could be a contribution to the community if open-sourced.


**Weakness**
* ***The novelty and the significance of the paper seem quite limited***. In particular, the paper targets a very specific optimization problem (i.e., shelf space allocation) and the proposed SNMM method seems to be a very straightforward generalization of the NMM method (by adding a scaler parameter) which is not convincingly justified with empirical evidence. It is crucial to discuss how the proposed method generalizes beyond the specific problem formulation and how significant it is. See “Novelty & Significance” in the next section for details.
* ***The presentation of the paper is not very clear or easy to follow***. In particular, the specific optimization problem considered, the key issue that the proposed SNMM aims to address and the significance of SNMM need to be more clearly presented. There are also some unjustified claims and missing mathematical definitions that need to be clarified. See “Clarity & Quality” in the next section for details.
* ***The experimental evaluation is not reproducible or convincing***. There are a lot of missing experimental details (even missing introduction of baselines) in the paper and no further experimental details are included for reproducibility. See “Reproducibility” in the next section for details.
* ***Potential risk of plagiarism***. In Sec 5.1.1-5.1.3, the starting sentences for introducing the datasets are all **copied** from [1], which may be considered a violation of academic integrity.

[1] Diminishing Returns Shape Constraints for Interpretability and Regularization.  Gupta et al. NeurIPS 2018.


**Summary Of The Paper:**

This paper considers a real-world problem of ​​shelf space allocation for retail stores using DNN models. It focuses on the selection of predictive model families considering the particular convexity constraint for the specific problem formulation. It then proposes a Scaled Neural Multiplicative Model (SNMM) which aims to improve model expressivity under the constraints and is empirically demonstrated on both synthetic and real-world datasets.

**Summary Of The Review:**

In summary, the paper has some crucial flaws: 1) lack of novelty and significance;  2) unclear presentation and lack of soundness 3) reproducibility concerns and potential violation of academic integrity. Thus I recommend a ‘strong reject’ for the paper.

---

> ### Author Response · Authors · 2022-11-18
> **Response to reviewer 1jyf**
>
> We thank the reviewer for the review. While there are some valuable suggestions, we respectfully disagree with overall assessment and in particularly on novelty & significance. As noted by other two reviewers the solutions significance is the ability to solve an otherwise intractable problem. We would like to address few of the questions below:
> >Potential risk of plagiarism. In Sec 5.1.1-5.1.3, the starting sentences for introducing the datasets are all copied from [1], which may be considered a violation of academic integrity.
>
> We strongly disagree with this observation. What we stated in those parts are merely artifacts describing the results & data sets and we did quote the reference.
> >The formulation targets a very specific optimization problem, which significantly limits the scope of the paper[...] or 2) incorporating a discussion of how the proposed method generalizes beyond the specific problem formulation.
>
> While the core illustration of the formulation was a specific business need for any retail, the formulation can be easily extended to other space planning needs: Revenue Optimization;Price Optimization ;Assortment Optimization
> >The proposed SNMM simply modifies the NMM by adding a scaler parameter for the input feature. This modification, however, seems to be minor and marginal. Furthermore, there may be many other ways that could be considered, e.g., by utilizing the composition of (monotonically increasing) convex functions, but are not discussed or compared in the paper. It is crucial to justify the significance of the proposed SNMM compared with potential variants
>
> We believe the simplicity of the subtle modification makes an otherwise intractable optimization to a tractable one. What this means in a real world scenario is the ability to scale the solution to a large and varying data sets. It certainly is possible there may be other variants, and evaluating them is an exhaustive work in itself, but it is beyond the scope of this paper.
>
> >In Eq. (9b), the purpose for introducing the max operator is not explained.
> >In Sec 4.4, how the optimization problem is solved is not discussed.
>
> While some of these observations may marginally improve some aspects of the paper, we believe the explanations and clarity are provided.
>
> >Some statements are not well explained or justified. Here are a few:
> >In Eq. (8), the two linear layers are composed without any linearity which still falls into the linear family. How does it increase the expressivity as claimed above?
> >At the start of Sec 3.2, the two reasons for the advantages of multiplicative models over linear models are not sufficiently supported. More explanations or some references are needed.
>
> In Eq. (8) by adding the two linear layers, the functional form of the model leads to a power cone formulation (convex optimization) that is easier to interpret and solve.
> In section 3.2, we show the advantages of a multiplicative model over a linear model when calculating the parameters for the objective function in 4.4 and highlight them in section 4.1 (Data Sparsity). The multiplicative nature of the model allows us to estimate the total effect of all other features despite data sparsity.
> >Missing mathematical definitions. Here are a few: In Sec 3.1, the equation for Fθ(x), both ϕ and P are not defined;In Eq. (3), the notation ϕp(wip) is confusing to me, what doe p mean as sub-script and super-script respectively? ;In Eq. (14), the notation N is not defined.
>
> We respectfully submit that these definitions are provided and are self explanatory
> >The reproducibility of the experimental results is poor. In particular, in Sec 5.1, the experimental setups (e.g., how the models are trained and how the optimization problem is solved) and even the baselines considered in Table 1 & 2 are not introduced. In Sec 5.2, the dataset used seems not publicly available, but the dataset descriptions are not detailed enough for reproducibility. There are no further experimental details included for reproducing the results in the paper.
>
> The core problem of shelf space to category of items is pivotal to many retailers, gathering and maintaining such data is highly proprietary. These datasets are unfortunately not available publicly. Creating a synthetic data set may be an option, but is extremely challenging. Further, we believe the framework laid out in the paper is comprehensive enough.

---

> > ### Comment · Reviewer_1jyf · 2022-11-21
> > **Further Suggestions & Comments**
> >
> > I thank the reviewers for their responses to my questions. After reading the authors' responses and other reviewers' feedback, I have a better grasp of the technical contribution of the paper and have accordingly adjusted the score for it. Here are my further comments and suggestions:
> >
> > * **Quoted sentences**: Although the quoted sentences are for describing datasets and experimental setups, I don't think it is valid to copy them from another paper directly. I hope this message is clearly conveyed, and I thank the authors for adjusting the quoted sentences in the new version.
> > * **Technical contributions**: I think the paper could be much improved by clearly emphasizing its motivations (the issue that the proposed SNMM aims to address) and technical contributions (compared with previous work or other simple variants), crucially *with supported claims or empirical evidence*. In particular, the lack of discussion of how SNMM breaks the expressivity bottleneck and considerations of other simple variants is not addressed in the authors' response, but to me is very important for justifying the significance of the proposed SNMM.
> > * **Empirical evaluation**: Given that the goal of the proposed SNMM is to handle the shape constraints to address the intractability of the optimization problem, the empirical evaluation should be focused on tractability and its potential tradeoff between expressivity. In particular, my suggestions would be:
> >   * *Rethinking the evaluation metric*: The MSE or other related metrics used in the paper do not reflect the tractability of the optimization problem, which blurs the benefit of the proposed method. The authors might consider evaluating the accuracy and/or time for solving the induced optimization problem.
> >   * *Including other baselines*: I agree with other reviewers that the other baselines should be included and compared.
> >   * *Improving the organization*: I would also recommend the authors improve the organization of the experimental section, in particular being clear about 1) how the experiments are designed to study a particular question related to the proposed method (e.g., the tractability compared to baseline methods, the improved expressivity of SNMM); 2) how the empirical evidence supports the claims made in the paper; 3) empirical details for reproducibility, which is very important given the dataset is not released.
> >
> > * **Clarity of the paper**: I would like to clarify that my concerns about the clarity of the paper including those about unjustified statements and missing definitions are my sincere feedback instead of criticism of the paper, which however seems not being respectively treated in the authors' response. I hope the authors would consider improving the writing for these parts, especially being more careful about introducing new notations (with clear definitions) and claiming some statements (with either empirical evidence or references).

---

### Official Review · Reviewer_AqEP · 2022-10-24

**Confidence:** 3
**Correctness:** 3
**Technical Novelty And Significance:** 3
**Empirical Novelty And Significance:** 4
**Recommendation:** 5

**Clarity, Quality, Novelty And Reproducibility:**

Clarity/quality/novelty are good.

By nature, the main experimental results of this work are not reproducible because they use a proprietary dataset. I don't think this is necessarily a reason for rejection, but it is a bit concerning given that the real-world data is where the benefit of the method is observed.

**Strength And Weaknesses:**

Strength:
* The paper studies an interesting a relevant applied problem. The problem is well explained and familiar to any reader, which (in my opinion) makes the paper interesting/fun to read. The proposed method is explained well and is novel to the best of my knowledge.
* The authors study a novel combination of neural networks and GAMs for effectively imposing shape constrained that are needed to model retail space allocation.
* The proposed technique performs well on the real-world datasets consistently.

Weakness:
* SNMM does not provide a consistent performance benefit, though it seems to improve performance in some cases. The validation MSE on the puzzle sales dataset is especially concerning, as it seems to be the worst of the considered methods. A similar result is observed on the wine dataset. It seems like a deeper discussion of why SNMM performs so poorly is needed here, instead of claiming that the error is within the range of other methods.
* Most of the baselines that outperform SNMM on the smaller datasets (e.g., DNN, RTL, SCNN) are not actually considered in the real-world experiments where SNMM performs the best.

Small Comments:
* In related work, you use the acronym GAM without stating what this means.
* The discussion on data sparsity and the difficulty of modeling/recording retail behavior is useful and provides good context for solving the proposed problem.

Questions:
* What are a few examples of real world applications the authors think this approach could be extended to? It would be nice to maybe give a few examples in the text to give readers an idea.
* Is (3) seen as a formulation that is easier to explain/interpret in comparison to (2)?
* Do the changes in (9a)/(9b) solve the problem of reduced capacity mentioned at the beginning of the section
* Why are different baselines explored on real data (Sec 5.2) vs. synthetic/small problems (Sec 5.1)?

**Summary Of The Paper:**

This paper considers the problem of optimizing shelf space in retail stores (i.e., how many fixtures to assign to each of several brands within a category of products for sale) using DNNs. Finding a tractable optimization formulation for such a problem is non-trivial, but the authors propose the scaled neural multiplicative model (SNMM) with shape constraints that leads to a tractable optimization problem. The authors evaluate this formulation empirically across a variety of synthetic/smaller datasets, then perform a case study on real-world data. Although results for SNMM are inconsistent on smaller/synthetic datasets, it seems to provide a performance benefit on real-world data relative to several baseline techniques.


**Summary Of The Review:**

To begin, I want to emphasize that this review reflects my initial impression of the work, and that I am completely open to further discussion with other reviewers/authors. My final score will mostly be based upon the subsequent discussion.

Currently, my main concern with the work are:
* Benefits are only observed on real-world datasets, while on the smaller/synthetic data the proposed method is actually quite a bit worse.
* No substantial discussion is provided regarding why the method performs so poorly compared to baselines on some datasets.
* Baselines on real-world data (where we see the benefit) are different from the smaller/synthetic experiments. The methods that outperform SNMM actually are not considered in the real-world case.

I think the work could be greatly improved by performing further analysis on the cases where SNMM performs poorly and providing a better understanding of why the real-world case is so different (plus explaining the problem with the selected baselines). Overall, the results on the real-world dataset are quite promising, and I believe this paper can be valuable/useful given further discussion/modifications.

---

> ### Author Response · Authors · 2022-11-17
> **Response to Reviewer AqEP**
>
> We thank the reviewer for the feedback and would like to address some of the questions/clarifications below.
>
> >SNMM does not provide a consistent performance benefit, though it seems to improve performance in some cases. The validation MSE on the puzzle sales dataset is especially concerning, as it seems to be the worst of the considered methods. A similar result is observed on the wine dataset. It seems like a deeper discussion of why SNMM performs so poorly is needed here, instead of claiming that the error is within the range of other methods.
>
> SNMM is a class of models that can handle shape constraints on the features, the performance of the SNMM model on the open-source dataset is not the major goal here. It is to provide an alternative easy to work with class of models to handle different types of shape constraints. We make no big claims on our method being SOTA. The purpose of the experiment on open-source dataset is to show that SNMM can handle different types of shape constraints.
>
> >Most of the baselines that outperform SNMM on the smaller datasets (e.g., DNN, RTL, SCNN) are not actually considered in the real-world experiments where SNMM performs the best.
>
> We did not consider these baselines since they do not have open-source code to reproduce the results. As noted in our experiment section, the results are taken from the referenced paper and we present SNMM as a comparable model.
>
> Further, although there exist different types of models that can handle convexity, SNMM is simple to understand and enforce constraints (convex, concave or monotone). Moreover, the optimization problem based on SNMM model will lead to a well-known power cone formulation and thus can be solved efficiently.
>
> >What are a few examples of real world applications the authors think this approach could be extended to? It would be nice to maybe give a few examples in the text to give readers an idea.
>
>  To name a few real world applications, the SNMM can be extended to
> 1. Revenue Optimization
> 2. Price Optimization (where pricing decisions must be made)
> 3. Assortment Optimization (model sales as a function of product location on the shelf, orientation, etc.)
> 4. Click-through-rate (CTR) maximization via spend allocation among different Ad. Channels (CTR to spend on a particular channel like Facebook or Google Ads can be modelled as a concave function)
> 5. Production planning applications where concavity of learning models yield tractable convex optimization (family of work related to clearing functions)
> 6. Network delay optimization (it is known that the delay in a network is a concave function of the amount of data packets sent via network) Network delay optimization identifies path for data to be sent with minimal communication delay
>
> >Is (3) seen as a formulation that is easier to explain/interpret in comparison to (2)?
>
> (3) is a much easier model to constrain the entire function to be concave or convex in comparison to (2). Further (2) leads to monomial or posynomials in the objective function that can be solved via geometric programming which is a generic non-linear optimization whereas the (3) leads to power cone formulation (convex optimization) that is much easier to interpret. It’s merely our choice of modeling since different variables appear in an additive manner in (3)
>
> >Do the changes in (9a)/(9b) solve the problem of reduced capacity mentioned at the beginning of the section
>
> Yes. This is how we provide more expressive capability to SNMM in comparison to NMM variants presented in the tables 3, 4 and 5.
> The NMM-L and NMM-CL do not have these changes outlined in 9(a)/9(b) and they share the same architecture as that of SNMM.
>
> >Why are different baselines explored on real data (Sec 5.2) vs. synthetic/small problems (Sec 5.1)?
>
> In the real-data our goal was to explore simple and interpretable learning models (we started with linear and take one step at time to add complexity while not trading explainability). In sec 5.1 the results are intended to show that our proposed model can handle shape constraints (we do not claim SOTA). Further, there is no open-source implementation available for the shape constrained models presented in Sec 5.1.

---

> > ### Comment · Reviewer_AqEP · 2022-11-21
> > **Response to authors**
> >
> > I thank the authors for their time and careful responses to my questions. Most of my smaller questions have been answered, and I now feel that I have a better grasp on the paper.
> >
> > My main thoughts are as follows.
> > 1. It seems your method is posed as a simple/easy-to-use method that is open-sourced. It is not SOTA compared to other techniques, but it is accessible and can handle a wide number of problems/constraints. I think this could be better emphasized in your paper from a writing perspective, which would make the performance differences with baselines more understandable. I think emphasizing more clearly the benefits of your method (e.g., applicability to more problems, simplicity/availability, open-source, etc.) within the paper would be easy to add and very helpful in terms of motivating the technique.
> > 2. I think it is very important to implement the best baseline methodologies for comparison on the real-world data. Even if these methods are not open-sourced, these methods need to be compared to the proposed method so that we as readers know how much we "lose" in performance by adopting the simpler SNMM approach. It is fine for it to not be SOTA, but we need to understand how much performance we sacrifice in return for SNMM's ease of implementation.  Even if some of these other methods might be somewhat difficult to implement, certain teams may be completely willing to perform this implementation if the performance improvement is significant on real-world datasets.
> > 3. You say SNMM can also be solved more efficiently in comparison to other methods. You need to provide results in your experiments to back this up (which further emphasizes the need to obtain/create implementations of at least a few baseline techniques). If SNMM for example can be solved 100X faster than widely-used baselines, this is a huge benefit for your paper and can make the contribution much more noticeable/significant. Providing some extra elaboration on the efficiency of SNMM has the potential to be a very nice point of differentiation for your paper in my opinion (i.e., people can accept being a bit below SOTA if you're much faster!).
> >
> > In general, I am more optimistic about this paper than other reviewers (although I agree that improving writing would be a good idea and improve the paper a lot). However, I think (in its current state) the paper is below the bar of acceptance for ICLR. My recommendations for moving forward would be:
> > - Improve the writing/motivation of the paper to make it easier to understand (the other reviewers also make some useful comments on readability that may be helpful)
> > - Really emphasize why your method is important in the writing (see above for more) and support these points of differentiation throughout the paper (especially focusing on how your experimental results connect to each of these benefits).
> > - Implement some of the baselines so that we get an idea of how they compare on real data. (additionally, you can provide more discussion on these baselines -- if they are really hard to implement then explaining this in the paper is good support for the utility of SNMM).
> > - Do some analysis on the efficiency of SNMM and how it compares to other methods.
> >
> > I believe if the authors make these changes, the paper will be drastically improved and have good potential (also, please remove direct quotes from previous papers/resources just to be safe and avoid any considerations of plagiarism). I wish the authors best of luck moving forward and thank them for their time in crafting this submission!

---

### Official Review · Reviewer_fHjV · 2022-10-26

**Confidence:** 3
**Correctness:** 2
**Technical Novelty And Significance:** 2
**Empirical Novelty And Significance:** Not applicable
**Recommendation:** 3

**Clarity, Quality, Novelty And Reproducibility:**

The paper needs to be edited to focus on its main contribution. Section 3 and 4.2 can be completely eliminated. In particular, it's unclear why linear models are mentioned at all.

In table 1, 2, and 3, the authors compare their approach to DNN, SCNN, RTL, and Cal Lin. It would be useful to point to the papers that describe these models to enable third parties to reproduce these results.

In table 4, the comparison is made against various versions of GLM, NAM, and NMM. It would be useful to let the reader know where to get details on these versions. It would also be beneficial to explain why the comparison is not made against the same baseline as for the previous experiments.




**Strength And Weaknesses:**

Strengths:
* The paper proposes a model that can be used directly by a convex optimizer, such as cvxpy.

Weaknesses:
* It's not clear that the model actually performs well. On the puzzle dataset, its accuracy was second to last, and on the wine quality dataset the model was the least accurate. That said, the model does well on the car sales dataset and the private dataset.
* Since the paper proposes an end-to-end solution, it would be useful to validate how well the model works with the optimizer to find an optimal assignment of shelf space.
* Other approaches have been proposed to solve similar problems under the umbrella of differentiable optimization. The related work section should mention a few of them, and it would be useful to add one of these to the baselines.
* Two of the datasets used in the evaluation are unrealistically small, with less than 200 datapoints. One of the dataset is not public, which will prevent other researchers from replicating the results and comparing new approaches against SNMM. Is there really nothing better available ?


**Summary Of The Paper:**

This paper introduces a neural network architecture, called Scaled Neural Multiplicative Model (SNMM). This architecture was designed with two main objectives in mind: first, the model can be fitted properly to reflect actual sales data, even though this data tends to be scarce and sparse. Second, the model is concave, which ensures it can be used by an off-the-shelf optimizer such as cvxpy to find the value of a subset of the input features that maximizes the outcome. Based on this model, the paper defines the problem of finding a shelf space assignment that maximizes sales. The paper evaluates the accuracy of SNMM on 3 public datasets, car sales, puzzle sales, and wine quality, as well as a private datasets of sales for an unspecified retailer.

**Summary Of The Review:**

The contribution of this paper is fairly small: it modifies the neural multiplicative model to ensure its concavity by using the sigmoid trick. Furthermore, the evaluation of the new architecture is limited, and does not convincingly demonstrate that the model works well in practice.

---

> ### Author Response · Authors · 2022-11-17
> **Response to Reviewer fHjV**
>
> Thank you for the detailed and helpful review. We address individual points below.
>
> > It's not clear that the model actually performs well. On the puzzle dataset, its accuracy was second to last, and on the wine quality dataset the model was the least accurate. That said, the model does well on the car sales dataset and the private dataset.
>
> The experiments demonstrate that the SNMM class of models can handle shape constraints on features and produce reasonable results. This was our goal. We agree that our car sales model was more accurate relative to other models than our puzzle or wine quality models. The assumptions we made regarding convexity and concavity led to less accurate models in certain cases. We have added text speaking to this point in section 5.1.2.
>
> > Since the paper proposes an end-to-end solution, it would be useful to validate how well the model works with the optimizer to find an optimal assignment of shelf space.
>
> We are able to relatively rapidly find optimal solutions to the shelf space assignment problem, a key requirement of our work. Validation of the predictive model and our work overall will come from sales figures during experiments involving changing layouts in stores. These experiments have not been completed.
>
> > Other approaches have been proposed to solve similar problems under the umbrella of differentiable optimization. The related work section should mention a few of them, and it would be useful to add one of these to the baselines.
>
> We have added a new sub-section on Differentiable Optimization to the end of the Related Work section of our manuscript. We will not be able to run experiments comparing this approach to our own in time to include in our revision. It’s worth noting that in our case there are benefits to developing an accurate predictive model independent of its use in the optimization, for example for use in communication with business stakeholders.
>
> > Two of the datasets used in the evaluation are unrealistically small, with less than 200 datapoints. One of the dataset is not public, which will prevent other researchers from replicating the results and comparing new approaches against SNMM. Is there really nothing better available?
>
> The goal of the experiments was just to demonstrate that SNMM can accommodate different assumptions regarding convexity and concavity. The datasets used in the experiments generating the results shown in Tables 1, 2, and 3 are the three public datasets used in the most relevant paper on shape-constrained models (Gupta et al. 2018). We can be reasonably confident that it makes sense to assume relations like diminishing returns to scale when modeling these data.
>
> > The paper needs to be edited to focus on its main contribution. Section 3 and 4.2 can be completely eliminated. In particular, it's unclear why linear models are mentioned at all.
>
> Sections 3 and 4.2 discuss the use of different predictive model forms within an optimization problem formulation. The linear case is commonly used in optimization; linear programming is a well-known, fast, and widely applied form of optimization. It is introduced here as a baseline. We have shortened our description of linear models in Section 3 to a single brief paragraph. We likewise shortened section 4.2. But we think it is useful to keep at least a few notes on linear models.
>
> > In table 1, 2, and 3, the authors compare their approach to DNN, SCNN, RTL, and Cal Lin. It would be useful to point to the papers that describe these models to enable third parties to reproduce these results.
>
> We have added text at the start of Section 5 referencing the papers which describe these models. The results regarding their performance originally appeared in the Gupta et al. (2018) paper.
>
> > In table 4, the comparison is made against various versions of GLM, NAM, and NMM. It would be useful to let the reader know where to get details on these versions. It would also be beneficial to explain why the comparison is not made against the same baseline as for the previous experiments.
>
> The text in Sections 4.2 and 4.3 describes the GLM, NAM, and NMM models tested. The results presented earlier regarding the performance of the DNN, SCNN, RTL, and Cal Lin models were taken from the Gupta et al. (2018) paper. We do not have results regarding the performance of these model classes on our data as there is no open-source code available to generate the results.

---

> > ### Comment · Reviewer_fHjV · 2022-11-21
> > **Response to authors**
> >
> > Thanks you for answering my questions. Looking at all the feedback, its seems that the main contributions of your paper are:
> > * introduce an easier to use method to solve the shelf assignment problem, as well as other important applications. Emphasizing more clearly these benefits in the paper and demonstrating them more convincingly in the experiment section would strengthen your paper.
> > * you also mention efficiency. If quickly solving the optimization problem is a strength of your approach, I recommend that you mention this explicitly in the paper and empirically compare the time to optimal solution of various solutions.

---

### Decision · Program_Chairs · 2023-01-20

**Decision:**

Reject

**Justification For Why Not Higher Score:**

See above.

**Justification For Why Not Lower Score:**

N/A

**Metareview: Summary, Strengths And Weaknesses:**

This paper presents a variant on the neural multiplicative model which can be used for demand modeling, and applies the method to shelf space allocation. The reviewers feel like there might be value in introducing this problem formulation if it were being released as a public benchmark that others could build on. However, the reviewers have a number of significant objections, including: (1) limited novelty, (2) clarity of both the problem formulation and the method, (3) the lack of an apparent improvement (it seems to underperform various baselines), and (4) the lack of reproducibility due to the lack of comparisons on publicly available datasets. I hope that the authors take the reviewers' comments into account in resubmission.